# Planting Date of Cotton in the Brazilian Cerrado Drives Boll Weevil (Coleoptera: Curculionidae) Infestation

**DOI:** 10.3390/insects14070599

**Published:** 2023-07-02

**Authors:** Patrícia J. Santos, Antônio M. Dias, Karolayne L. Campos, Ana C. A. Araújo, Andréa A. S. Oliveira, Fábio A. Suinaga, Jorge B. Torres, Cristina S. Bastos

**Affiliations:** 1Faculdade de Agronomia e Medicina Veterinária (FAV), Campus Darcy Ribeiro, Universidade de Brasília (UnB), ICC-Sul, Asa Norte, Brasília 70910-900, Distrito Federal, Brazil; patriciajesusdossantos@gmail.com (P.J.S.); antoniokaimbe@gmail.com (A.M.D.); anacalvesaraujo@gmail.com (A.C.A.A.); cschetino@unb.br (C.S.B.); 2DEPA-Entomologia, Universidade Federal Rural do Pernambuco (UFRPE), Rua Dom Manoel Medeiros s/n, Dois Irmão, Recife 52171-900, Pernambuco, Brazil; karolaynelopescampos@gmail.com (K.L.C.); jorge.torres@ufrpe.br (J.B.T.); 3Departamento de Entomologia, Campus Universitário, Universidade Federal de Viçosa (UFV), s/n, Viçosa 36570-000, Minas Gerais, Brazil; andreasantos1695@gmail.com; 4Embrapa Hortaliças (CNPH), Rodovia BR 060 Km 9—Samambaia Norte, Brasília 70351-970, Distrito Federal, Brazil

**Keywords:** *Anthonomus grandis grandis*, *Gossypium hirsutum*, cultural control, preventive tactics, cotton IPM

## Abstract

**Simple Summary:**

Cotton is cultivated in subtropical and seasonally dry tropical areas in the northern and southern hemispheres as the most common natural fiber used to make textiles. In the Americas, especially Central and South America, cotton is infested by a beetle named the boll weevil. This weevil feeds directly on the cotton reproductive structures, causing up to 70% yield losses. The flower buds and bolls damaged by weevils fall on the ground, becoming lost or are retained by the plants producing poor-quality fiber. Boll weevil is a target of 19–25 insecticide applications of broad-spectrum insecticides throughout the season, harming non-target organisms and growers. Therefore, alternatives to manage boll weevil need to be pursued. In this work, we studied how the planting dates of cotton could negatively affect boll weevil infestation. We studied three contrasting planting dates, monthly spaced, and looked at how the cotton plant and the boll weevil responded. We noticed that the middle-planting date was unfavorable to the boll weevil compared to the early and late-planting dates because cotton plants developed faster then and so escaped from pest infestation. Thus, cotton crop seeded late was more prone to receive numerous and fertile insects dispersing from surrounding cultivations while early planted cotton was more prone to receive dispersing adults from the refuge areas avid for infesting cotton reproductive structures.

**Abstract:**

Although the boll weevil (BW), *Anthonomus grandis grandis* (Coleoptera: Curculionidae) has been attributed to the significant losses caused to cotton yield in the Americas, the categorization as a quarentenary pest in places where it is still not occurring has increased its relevance worldwide. In areas where it is widespread, pest suppression relies on many broad-spectrum insecticide applications. However, other control tactics are sought. Considering that early-flowering cultivars escape from boll weevil infestation, we investigated if three different planting dates (November, December, and January) could alter the plant life cycle, allowing the plants to escape from boll weevil infestation. Field trials were run in two seasons (2014/2015 and 2017/2018), and variables (days required to reach each flowering stage, fruiting plant structures—undamaged and damaged by the BW, and totals—number of boll weevils on plants and that had emerged from fallen structures on the ground) were assessed over 29 and 33 weeks, respectively. Based on the number of days required to initiate and terminate the flowering stage, the time to reach the economic threshold (ET), the number of undamaged, damaged, and the total reproductive structures, we concluded that planting dates in December for the Central Cerrado of Brazil should be preferred over the other two tested dates. Cultivations run at this planting date, anticipating the flowering period initiation and termination, reduced infested flowering structures, and delayed the decision making to control the pest, when compared to the other two planting dates.

## 1. Introduction

Cotton planting dates may affect pest infestation directly by altering the pest life cycle [1] and indirectly by phenologically escaping the peak of infestation. The boll weevil (*Anthonomus grandis grandis* Boh., Coleoptera: Curculionidae) dispersal from refuge areas [2,3] and from surroundings, currently or previously cultivated fields [4], might be affected when modifying plant growth by increasing [4,5] or reducing [4,6] suitable reproductive structures for infestation. This is a commonly used practice adopted while using precocious types of cotton [7], delayed [8] or anticipated [9] synchronized cultivation, and the fallow period [10]—all of them aiming at escaping from boll weevil infestation. In many situations, when precocious cotton is unavailable or does not contribute to significantly reducing the pest populations, planting dates work by offering cotton growers the option to escape high pressure situations caused by pest infestations [11]. Depending on the planting date, there is a phenological asynchrony between pest occurrence and plant development caused by shorter and synchronized flowering stages [6,12]. On the other hand, the synchrony between preferred reproductive structures and pest occurrence favors pest outbreaks [4,5], a significant issue regarding boll weevil infestations.

Boll weevil is the main pest of cotton in the Americas and a continuous threat to other cotton-growing areas [13]. The substantial economic losses caused, and the challenges faced by growers to achieve its control, categorize boll weevil as an A1 quarantine pest by the EPPO [13] or a quarantine pest in other countries and continents [14]. Boll weevil adult males and females rely on the cotton reproductive structures (buds, flowers, and bolls) for food and reproduction. All developmental stages of the boll weevil occur inside these fruiting structures [15,16] partially protected from mortality factors. Adults can also feed on plant terminals mainly during the absence of cotton fruiting structures [15]. The infested reproductive structures are retained by the plants or fall on the ground [10,17], and the emerging adults initiate a new generation cycle [18]. Therefore, reproductive structures damaged are completely (fallen buds and young bolls up to ≈2 week after formation) [15,19] or partially (mature bolls retained by the plants) lost [20], causing up to 70% of yield reduction [21].

Integrated or multiple control measures preconized by integrated pest management are not commonly adopted in boll weevil management. With the frequent outbreaks, synthetic insecticide applications are often used against boll weevils [22] with an average of 19.6 applications per season [23]. Although often used, none of the insecticides sprayed to control the boll weevil impact the pest inside the reproductive structures [24], where they undergo all their immature developmental stages [25]. Furthermore, adult weevils gain some protection from direct contact with applied insecticides by the bracts that enclose the reproductive structures, while the exposure to sub-lethal doses reaching the insects does not restrain population build-up and may contribute to the selection of resistant populations [24]. The use of other control measures that impair population build-up may contribute to precluding them from reaching the economic threshold (ET), reducing the need for chemical control intervention in accordance with IPM principles [26].

In many tropical and subtropical countries, cotton planting dates can be variable because there are no environmental restraints to grow the plants, expanding the cultivation windows [2,27,28]. In addition, many of these cultivation areas lack restrictions for population maintenance during the off-season. Although the boll weevils reproduce in cotton [29], they can survive feeding on a diverse food diet [30], and on cotton regrowth [31]. Then, under environmental conditions that are not harsh enough to cause a significant decline in the populations [32], adults will be maintained during the off-season. Hence, as soon as cotton plants are available for infestation again, dispersing adults will reinfest the plants from the refuge areas [32] or from surviving individuals in the area [33]. The existence of extensively cultivated areas established in large planting date windows may favor population build-up across the fields [34,35]. Therefore, planting dates can have an impact on pest outbreaks. Based on that, this study assessed the effects of three planting dates during two growing seasons on cotton boll weevil infestation. We hypothesized that early-planted cotton would be less infested than late-planted cotton, concurring with the production of a smaller number of reinfesting adults and, consequently, with infested reproductive structures that would require fewer insecticide applications.

## 2. Materials and Methods

### 2.1. General Conditions

Field experiments were run during two growing seasons, ca., 2014/2015 and 2017/2018. Environmental conditions prevalent during the growing seasons are presented in Figure 1. The fields were located in the Fazenda Água Limpa (FAL) belonging to the University of Brasília, Vargem Bonita, Brasília, Distrito Federal, Brazil, 15.98 and 47.97 latitude and longitude, respectively, and 1080 m a.s.l. The experimental area was fertilized with a mixture of cow and sheep manure applied at a rate of 20 tons ha^1^ directly before planting and 20 tons ha^1^ 30 days after sowing. Additionally, natural phosphate (Yoorin^®^, Poços de Caldas, Minas Gerais, Brazil) comprised of 18% P_2_O_5_, 18% Ca, 7% Mg, and 10% Si was applied mixed with manure before planting at 1750 kg ha^−1^. Insecticide applications were not adopted during the experimentation period. Insecticide applications would affect boll weevil colonization and, hence, population growth masking the contribution of the studied planting dates. Weeds were controlled with hand hoes and water demand was supplied using sprinkler irrigation when rain did not occur for three consecutive days during the critical developmental stages (seedling, flowering, and boll initiation). Soil cultivation followed the procedures for growing cotton in Brazil [36]. The tips of cotton plants (≈30 cm) were manually removed with the aid of pruning shears (Tramontina^®^, Carlos Barbosa, Rio Grande do Sul, Brazil) after the plants had reached ≈1.7 m tall or when internodes had exceeded 4 cm. Planting rows were spaced of 0.9 m and seed density was 10 plants m^−1^. Cotton cultivars, both having white lint, were FMT 705 and BRS 293 for 2014/2015 and 2017/2018 seasons, respectively. The treatments were three planting dates: 20 November and 16 December 2014, and 20 January 2015, corresponding to the planting dates in the 2014/2015 season; 21 November and 20 December 2017, and 22 January 2018, which corresponded to the planting dates in the 2017/2018 season. The experimental design was a randomized block with four replicates per treatment. The treatments (planting dates) were variable over time; thus, those plots randomly assigned to 2nd- and 3rd- plantings dates were kept free from weed infestation with the aid of transparent PVC films that were removed by the time that sowing occurred. In the 2014/2015 season, the total experimental area had 1656 m^2^ (138 × 12 m) with each plot measuring 12 × 11 m (L × W). In the 2017/2018 season, the total experimental area had 1550 m^2^ (62 × 25 m) and each plot measured 20 × 5.5 m (L × W). The plots were separated from each other by 1 m width buffer areas, in order to keep a regular distance between them. 

### 2.2. Variables

The number of days that cotton plants needed to produce the first bud, white flower, bolls, and open bolls was determined by examining 10 plants per plot randomly assigned during the first evaluation, located on the central part of the plots, which were marked for tracking the plant phenology. Upon the detection of buds of ≈5 mm diameter, evaluations of plants and fallen structures began and were maintained at weekly intervals. Ten plants randomly selected and located in the central part of the plots were evaluated per plot and all the abscised reproductive structures in each plot were collected, stored in 5 L capacity plastic bags, and taken to the laboratory for further evaluations. Evaluations were maintained for 33 weeks (from 75 to 298 days after planting, DAP) during the 2014/2015 season, and for 29 weeks (from 69 until 274 DAP) during the 2017/2018 season.

The total numbers of reproductive structures undamaged (squares, flowers, and bolls), reproductive structures damaged (showing boll weevil feeding and oviposition punctures and non-opened damaged bolls), and total reproductive structures were assessed on the plants and on abscised structures. Boll weevil adults found on the plants and reproductive structures found on the ground with exit openings were also recorded. After the evaluation, abscised reproductive structures were placed into transparent 5 L plastic containers, with a 15 × 15 cm opening on the lid, covered with voile fabric to allow gas exchange, and maintained at lab prevailing conditions. Evaluations of these abscised structures were performed weekly until 21 d after collection, a time long enough for boll weevil immature stages to complete development and to reach adult stages [37]. Emerged adults were tallied weekly with counted weevils removed. After 21 days of evaluations, the fruiting structures were opened, and the numbers of adults and immatures found inside were recorded.

### 2.3. Statistical Analysis

The number of days required for the first visual reproductive structures on the plants was submitted to one-way analysis of variance [38] aiming at verifying the influence of the three planting dates on the reproductive phenological time. Percentages of infested reproductive structures across all sampling dates were calculated for plant and abscised fruiting structures by dividing the number of damaged ones by the total of the reproductive structures and multiplying by 100. Pair comparisons were performed between infested reproductive structures found on the plants and abscised (found on the ground) using a non-paired *t*-test [38]. Furthermore, the percentage of infested reproductive structures found on the plants was used to indicate the time that the infestation reached the economic threshold (ET), commonly used by cotton growers to manage boll weevil populations [39]. 

The numbers of total reproductive structures, undamaged structures, and damaged structures on cotton plants were reduced to the mean by the plant for all evaluations. The results were submitted to a one-way repeated measure analysis of variance with sampling weeks as a blocking factor within each season, using four replications of the three treatments (i.e., planting dates). Furthermore, seasonal means were obtained for total reproductive structures, undamaged structures, and damaged structures surveyed on the plants. The separation of the treatment means within each sampling week, and the overall seasonal means per treatment were performed by Tukey HSD test (α = 0.05). 

The same variables (total reproductive structures, undamaged structures, and damaged structures) were assessed for abscised structures together with the number of larvae and pupae and emerged boll weevil adults. These variables were summed across all the evaluation weeks and compared among the treatments by one-way analysis of variance [38]. 

## 3. Results

### 3.1. Time for Producing the First Reproductive Structures

During both seasons, cotton plants cultivated in December produced flower buds, white flowers, and young bolls earlier than when they were cultivated in November and January (Table 1). There was one exception of a similar time to produce young bolls by cotton plants cultivated in November and December during the 2017/2018 season (Table 1). Open bolls were produced earlier by cotton plants cultivated in December compared to plants cultivated in January and November in 2014/2015 and were produced at a similar time by the plants cultivated in November and December of 2017/2018 season and in less time than that required for plants cultivated in January (Table 1).

### 3.2. Infestation on Plant and Abscised Fruiting Structures 

Percentage fruiting structures damaged by the boll weevil reached the ET during the sampling weeks 1, 6, and 2 for the first, second, and third planting dates of 2014/2015 season, and remained above the ET for the following evaluations (Figure 2). In 2017/2018 season, the ET was reached in the sampling week 4 for the first and second planting dates, and in the sampling week 1 for the third planting date. The infestation remained above the ET until the last week of evaluation (Figure 3). The percentage of abscised fruiting structures damaged was significantly higher than the percentage of those damaged structures on the plants, except at 195 DAP for the second planting date of the 2014/2015 season (Figure 2), and at 174 and 274 DAP for the second and third planting dates, during the 2017/2018 season (Figure 3). 

The average percentages of fruiting structures damaged on the plants were 10.5 to 44.5, 4.7 to 35.4, and 1.7 to 45.9, and of abscised structures 64.2 to 91.9, 61.1 to 92.4, and 66.7 to 97.3, respectively, during the first, second, and third planting dates in the 2014/2015 season (Figure 2). In the 2017/2018 season, the average of percentages of fruiting structures damaged on the plants were 1.5 to 50.1, 0.5 to 43.9, and 10.8 to 57.5 and of abscised structures 33.9 to 93.2, 17.0 to 81.9, and 46.6 to 91.5, respectively, during the first, second, and third planting dates (Figure 3).

### 3.3. Reproductive Structures Total, Undamaged, and Damaged, and Boll Weevil Immature and Adult Numbers

The total of fruiting structures and undamaged structures produced by cotton plants cultivated in the first planting date was superior during the three first sampling weeks compared to the second and third planting dates, which did not differ between them over the 2014/2015 season (Figure 4a,b). The same result was observed with total fruiting structures for sampling weeks 23 and 24 and with undamaged fruiting structures for sampling week 18 (Figure 4a,b). The first planting date also produced the highest number of reproductive structures, total and undamaged, in the sampling weeks 4, 5, 7, and 8 that differed from both, the second and third planting dates, which were also different from each other; the same occurred in sampling week 6 for the total reproductive structures (Figure 4a,b). The total number of reproductive structures from sampling weeks 9 to 14, and sampling weeks 16 and 27, was the highest and did not differ between the first and second planting dates while both were different from the third planting date (Figure 4a). In the sampling weeks 15, 17, and 18, the total reproductive structures produced by the second planting date did not differ from both the first and third planting dates which had different numbers from each other (Figure 4a). The total, undamaged, and damaged numbers of fruiting structures was maximum in the third planting date during sampling weeks 32 to 37 and minimum in the first and second planting dates (Figure 4a–c). Similar results were observed for sampling weeks 6, 9, 10, and 13, when the first and second planting dates had a greater number of reproductive structures undamaged, which was different from the third planting date (Figure 4b). During the sampling weeks 15 and 29, the second planting date produced reproductive structures undamaged similar to the first and third planting dates, although they differed from each other (Figure 4b). Fruiting structures damaged per cotton plant were at a maximum in the first planting date of 2014/2015 season and a minimum in the second and third planting dates in the sampling weeks 4, 5, and 8 (Figure 4c). Similar and maximum numbers of reproductive structures damaged were found per plant in the first and second planting dates during sampling weeks 9, 10, 11, and 17, while the minimum numbers were produced by plants cultivated in the third planting date (Figure 4c). The first planting date produced the highest number of reproductive structures damaged during sampling week 12 and differed from both the second and third planting dates (Figure 4c). Cotton plants cultivated in the second planting date produced total fruiting structures that did not differ from the first and third planting date during sampling weeks 6, 7, 13, 15, 16, and 27, while they differed from each other (Figure 4c). Seasonal means of reproductive structures total and damaged, were at a maximum in the first planting date and a minimum in the second and third planting dates (Figure 4c).

In the 2017/2018 season, the total number of cotton fruiting structures and undamaged structures per plant were maximized in the first three weeks of the first planting date and reduced in both the second and third planting dates (Figure 5a,b). The total, undamaged and reproductive structures damaged, was higher during sampling weeks 7 to 10 for the first and second planting dates than in the third planting date (Figure 5a–c), and the same happened with reproductive structures undamaged in sampling week 5 (Figure 5b). A similar pattern of higher reproductive structures total, undamaged and damaged during the sampling weeks 22 and 24, and higher reproductive structures total and damaged in sampling week 13, was recorded in the second and third planting dates that differed from the first planting date, although not differing from each other (Figure 5a–c). The maximum number of total and reproductive structures undamaged was produced during sampling weeks 4 and 6 for the first planting date, which differed from both the second and third planting dates (Figure 5a,b). Likewise, total reproductive structures during sampling week 5 were superior for the first planting date (Figure 5a). The third planting date maximized the total, undamaged, and damaged numbers of reproductive structures from sampling weeks 25 to 28 and differed from both the first and second planting dates which had similar results (Figure 5a–c). The first planting date had the maximum number of reproductive structures damaged differing from both the second and third planting dates during sampling weeks 3 to 6 (Figure 5c). Additionally, the first planting date produced a similar number of reproductive structures damaged for the third planting date and lower than the second planting date during sampling week 20 (Figure 5c). The seasonal mean of reproductive structures damaged was maximized in the first planting date and minimized in the second and third planting dates (Figure 5c). Nevertheless, the total and undamaged numbers of reproductive structures were similar among the planting dates (Figure 5c).

Concerning the abscised structures, statistical differences (F_11, 6_= 6.43; *p* = 0.0322) were produced only for the total number of reproductive structures damaged in the 2014/2015 season, with averages of 18,890.25 ± 515.40, 16,097.25 ± 2114.45, and 13,391.25 ± 1632.37 for the third, first, and second planting dates, respectively. The second planting date did not differ from both, the first and third planting dates that differed from each other. In the 2017/2018 season, significant differences were observed for the number of boll weevil larvae (F_11, 6_= 13.81; *p* = 0.0057), when the greater number of larvae was observed in the third planting date (77.2 ± 10.95) and lower and statistically similar numbers were observed in both the first (15.5 ± 4.33) and second (28.7 ± 6.13) planting dates.

The total number of boll weevil adults found during plant sampling or emerged from abscised structures throughout the season did not differ in both seasons. In the 2014/2015 season, the numbers of adults (mean ± SE) found on the plants were 27.5 ± 4.17, 22.25 ± 9.59, and 22.75 ± 4.90 weevils; while the numbers of weevils that emerged from abscised structures were 6550 ± 1135.03, 5845.5 ± 926.21, and 6340.50 ± 152.47, for the first, second, and third planting dates, respectively. In the 2017/2018 season, the average numbers of adults found on the plants were 21.5 ± 9.85, 9.5 ± 2.87, and 12.5 ± 2.53 weevils; while the numbers of weevils emerged from abscised structures were 2124.5 ± 251.87, 2102.75 ± 27,152, and 231,225 ± 144.68, for the first, second, and third planting dates, respectively.

## 4. Discussion

Cotton is a row crop with a long cycle (ca. ≈150 days) compared to other row crops that share the same agro-ecosystem in the Brazilian Cerrado Central (e.g., soybean, corn, sunflower, etc.). The planting window in the Central Cerrado and the large area growing cotton in Brazil (close to 1 million hectares in 2022/2023 season), allow two rainfed row-crop cycles per year and up to three cycles when rotating the crop with a short-cycle crop in the first planting window. Thus, growers who decide to cultivate cotton in the first planting window (October–November) run one crop cycle per year, and growers who decide to cultivate cotton in the second and third planting windows take advantage of having an early crop cycle production with soybean or corn from October to January before seeding cotton. Thus, growers rely on cotton as a rainfed crop for the second cropping cycle. The cotton plants develop well up to April and early May with mature and open bolls during the dry periods of June and July with high-quality fiber. This cropping strategy has allowed high revenue per cultivated area. Despite that, the results presented here are the first to consider these planting dates in relation to boll weevil infestation, the most important pest of cotton in Brazil.

The results confirmed that planting dates affect cotton plant development and fruit structure production, consequently, boll weevil infestation. The time required for the emergence of the reproductive structures infested by the boll weevil was significantly reduced on the second planting date compared to the other two dates. It is known that plants require a specific amount of heat (degree-days) to develop from one point in their life cycle to another that can be delayed or advanced depending on the heat accumulated over time [40,41]. Considering that the second planting date was closer to the summer solstice in the southern hemisphere, it seems to have worked to advance cotton plant development caused by a faster accumulation of heat units [40,42]. This is especially desirable in pest management because plants can escape from pests when the time required to produce the structures preferred by infestation is accelerated; the same occurs when they exhibit short life cycles. The escape from infestation promoted by early flowering cotton cultivars is already known and described [7,9]. The escape from infestation is also observed when the plant life cycle is reduced based on the planting dates. The consideration of successive planting dates, as is common in many tropical and subtropical regions, is especially important for boll weevil management because boll weevil females fed on alternative food sources do not produce viable eggs [43], while females fed exclusively on preferred buds (5.5–8.0 mm diameter) show high fecundity [15] and the ability to increase 60× each generation [44]. Thus, in the case of the present study, the reduction in time required to initiate and terminate the flowering period in the second planting date seems to have avoided the infestation by dispersal of more numerous and more fertile adults coming from the first planting date or from the surroundings. On the other hand, cotton plants cultivated on the first and third planting dates exhibited delayed flowering periods compared to the second planting date. In the case of the third planting date, it was expected that this cultivation would receive the numerous and highly fertile adult weevils that originated in the previous crops. Therefore, the planting dates affected infestation promoted by the boll weevil on the cotton plants.

The impact on infestation was observed when the decision to manage boll weevil populations based on ET was delayed in the second planting date compared to the first and third planting dates during the 2014/2015 season and compared to the third planting date during the 2017/2018 season. In addition, the number of weeks in which the second planting date had maximum numbers of reproductive structures damaged was lower than the first and third planting dates for the 2014/2015 season and was similar to the first planting date and lower than the third planting date for the 2017/2018 season. Furthermore, the seasonal means indicate a lower number of fruiting reproductive structures damaged in the second and third planting dates in both seasons, although these dates also produced lower total reproductive structures. In addition, the third planting date had more abscised damaged structures and larvae from these structures. It seems that adults infesting the third planting date fed on a more nutritious diet favor the fecundity of adults and the survival of immatures [15,44], justifying the results found. On the other hand, adults infesting the plants cultivated on the first planting date and dispersing from refuge areas are less prone to reproduce [43], which must account for the delay in decision making, as well as for the production of similar numbers of reproductive structures damaged compared to the second planting date, over many weeks of evaluation. The second planting date resulted in advanced flowering period initiation and termination, likely escaping from the infestation of highly fertile weevils from the late emergence in the first planting date.

The variations perceived in the initial boll weevil infestation for the first planting date across seasons (2014/2015 and 2017/2018) were likely to have originated from the differential numbers of dispersing adults from the refuge areas. This hypothesis arose from the contrasting numbers of emerging adults from the abscised structures collected in the first planting date across seasons, which were ≈3× higher for the 2014/2015 season compared to the 2017/2018 season. There are many environmental conditions faced by weevils such as dryness [45], preceding and surrounding cultivated crops [35], and cultural practices adopted after cotton harvest [46] that severely impair survival during the off-season period and significantly reduce the numbers of adult weevils colonizing new cotton fields when a season begins. Therefore, when these conditions impair weevil survival and maintenance in the area, initial numbers of infesting adults are reduced resulting in lower numbers of infested reproductive structures and, as a consequence, in lower numbers of emerging adults from infested reproductive structures. This probably explains the differences in the initial infestation caused in cotton plants cultivated between seasons and should not be a surprise as climatic conditions were roughly similar at the beginning of each season.

The delay in the decision making to control a pest reduces the number of insecticide applications and environmental contamination [47,48,49]. In addition, it decreases the chances to select resistant populations, and contributes to natural enemy preservation as well as reducing the chances of having secondary pest outbreaks [47,48,49]. Therefore, planting dates that delay or reduce chemical control are highly desirable on cotton IPM. The decision to spray cotton fields against the boll weevil has been made, based on the sampling of infested reproductive structures found on cotton plants. We noticed that the percentage of infested structures found on cotton plants reached a maximum of ≈60% and oscillated in levels much lower than that and close to 30% during most of the sampling. However, the infestation of abscised structures was close to 100% and above 80% in most of the sampling weeks. It is a source of infestation pointed out by previous studies [36] that is not currently considered in the effort to control boll weevil populations. Therefore, abscised fruiting structures should be incorporated in future studies that deal with decision making to control the boll weevil. Additionally, although the number of adults that emerged from abscised reproductive structures was similar among the treatments, especially in the first season, the number of adults obtained from the second planting date was considerably lower (≈700 adults lower than the number obtained in the other two planting dates) and must justify the inclusion of this source of infestation on the decision-making process. The additional number of insects produced by other planting dates represents the greater colonization potential of the subsequently cultivated fields.

Based on the results, the second planting date must be preferred over the first and third planting dates tested while focusing on boll weevil management. In cases where the other two planting dates are practiced, it will be important to give special attention to the sampling procedures allowing early detection and rapid adoption of IPM tactics, especially for the third planting date, which is expected to receive highly fertile adult weevils dispersing from the previously cultivated fields and to present delayed reproductive phenology. Therefore, considering that the third planting date is our late-planted cotton, the first planting date is our early planting date, and the second planting date is our middle-planted cotton. We rejected our initial hypothesis that early-planted cotton is less infested than late-planted cotton. Indeed, cotton planted close to the summer solstice of the southern hemisphere in December or middle-planted cotton, which was cultivated during the second planting date, diminishes pest outbreaks and therefore, should be preferred over the other two planting dates, early and late-planted cotton cultivated in November and January, respectively.

## Figures and Tables

**Figure 1 insects-14-00599-f001:**
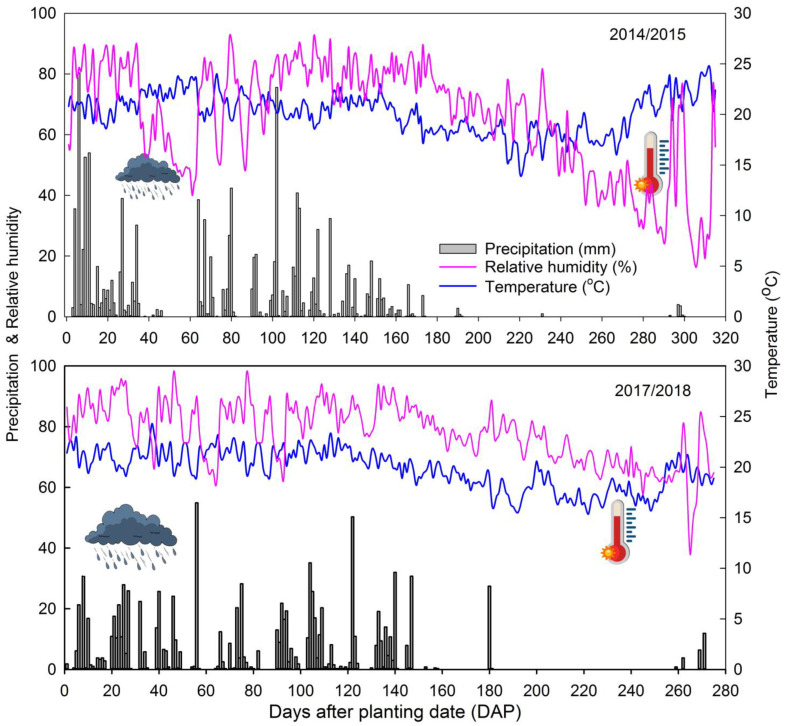
Mean of daily climatic conditions monitored during the growing seasons 2014/2015 and 2017/2018 in the location of the experiments.

**Figure 2 insects-14-00599-f002:**
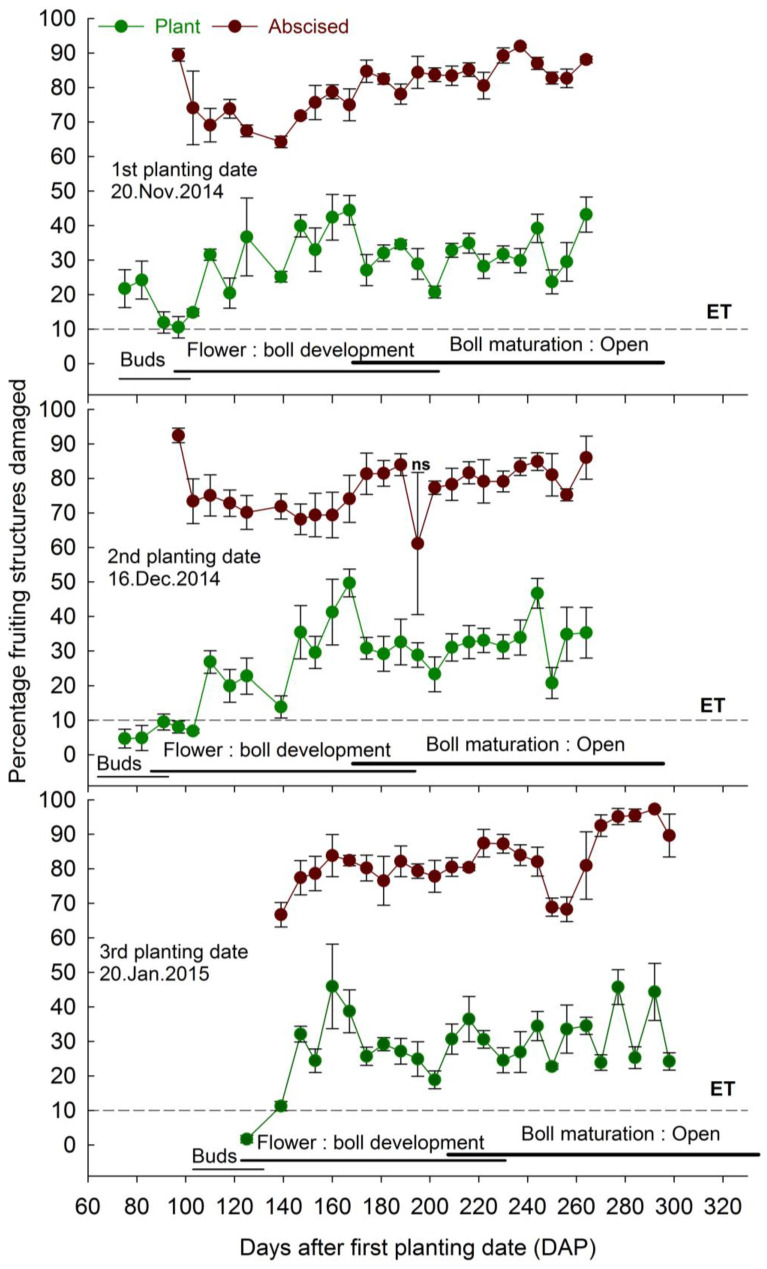
Percentage (means ± SE) of cotton fruiting structures damaged by cotton boll weevil recorded on plants and collected on the ground (abscised structures) as a function of days after the first cotton planting date (DAP) for the 2014/2015 season. Note: ET stands for economic threshold; all means were different between plant and abscised structures by non-paired *t*-test at *p* < 0.05, except where specified (ns).

**Figure 3 insects-14-00599-f003:**
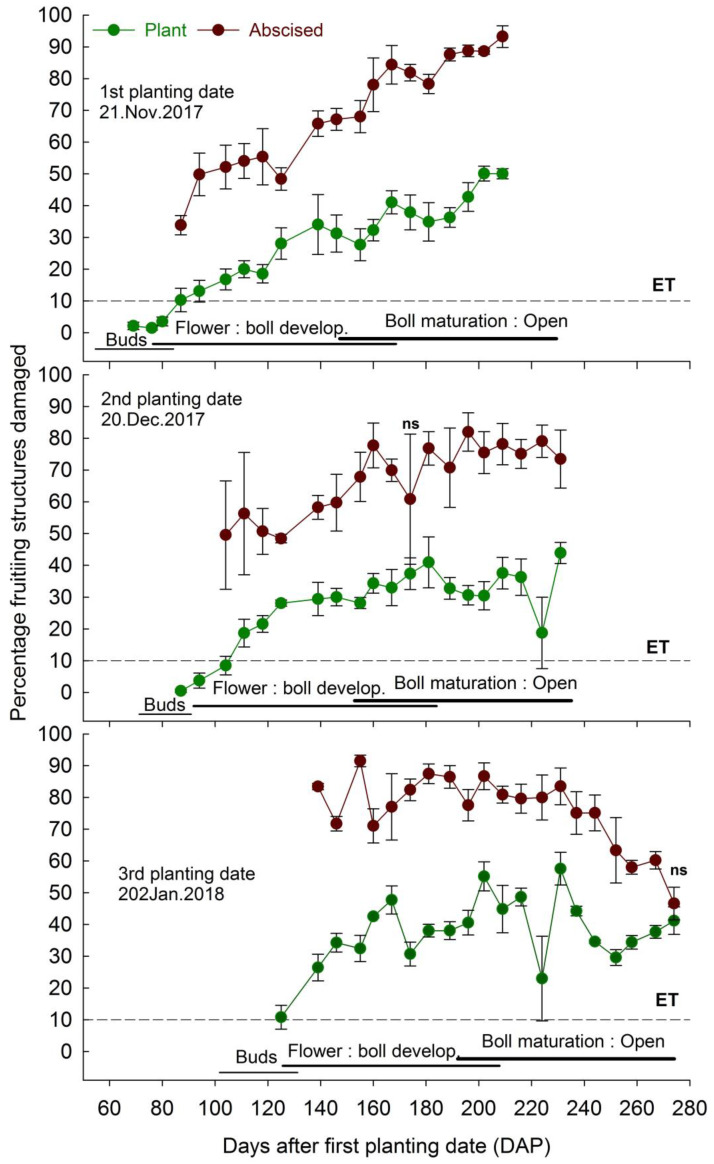
Percentage (means ± SE) of cotton fruiting structures damaged by cotton boll weevil recorded on plants and collected on the ground (abscised structures) as a function of days after the first cotton planting date (DAP) for the 2017/2018 season. Note: ET stands for economic threshold; all means were different between plant and abscised structures by non-paired *t*-test at *p* < 0.05, except where specified (ns).

**Figure 4 insects-14-00599-f004:**
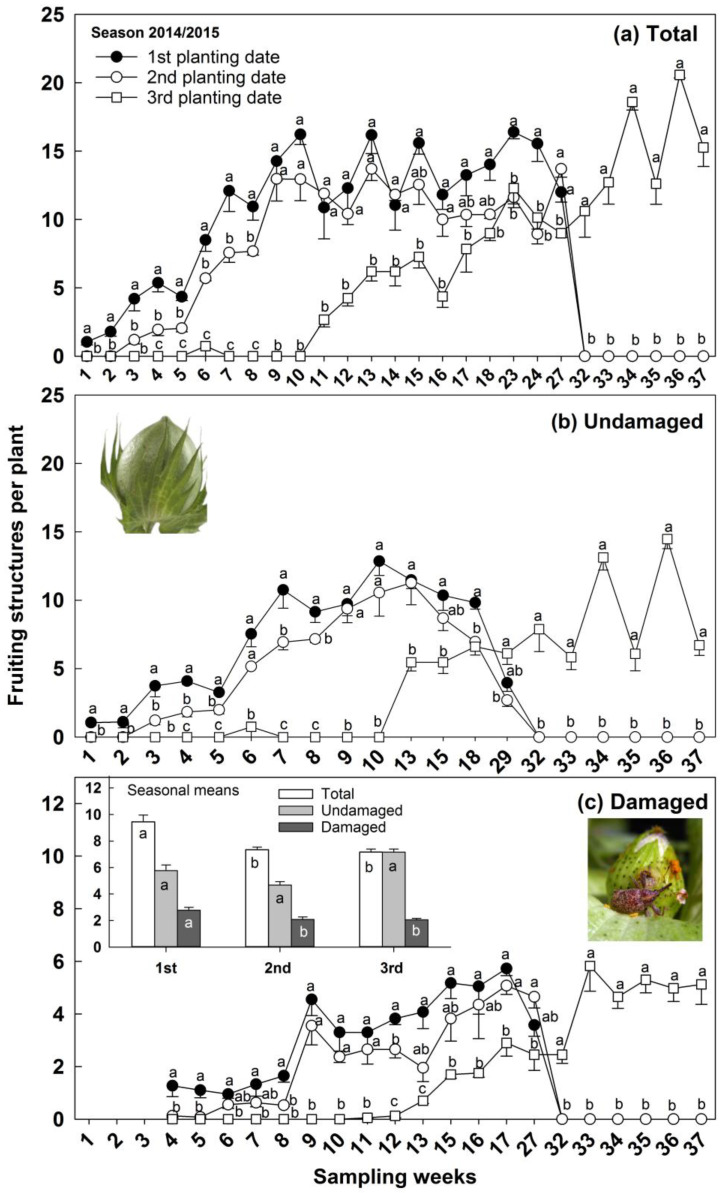
Number of cotton fruiting structures (mean -SE) per plant sampled over the season as a function of sampling weeks or seasonal mean (graphic inside) in 2014/2015 season. Symbols with different letters indicate statistical differences among planting dates within each sampling week, and the bar bearing different letters indicates differences for each fruiting structure category among sampling dates by Tukey HSD test (α = 0.05). Note: the scale in damaged structures is different to better display the outcome.

**Figure 5 insects-14-00599-f005:**
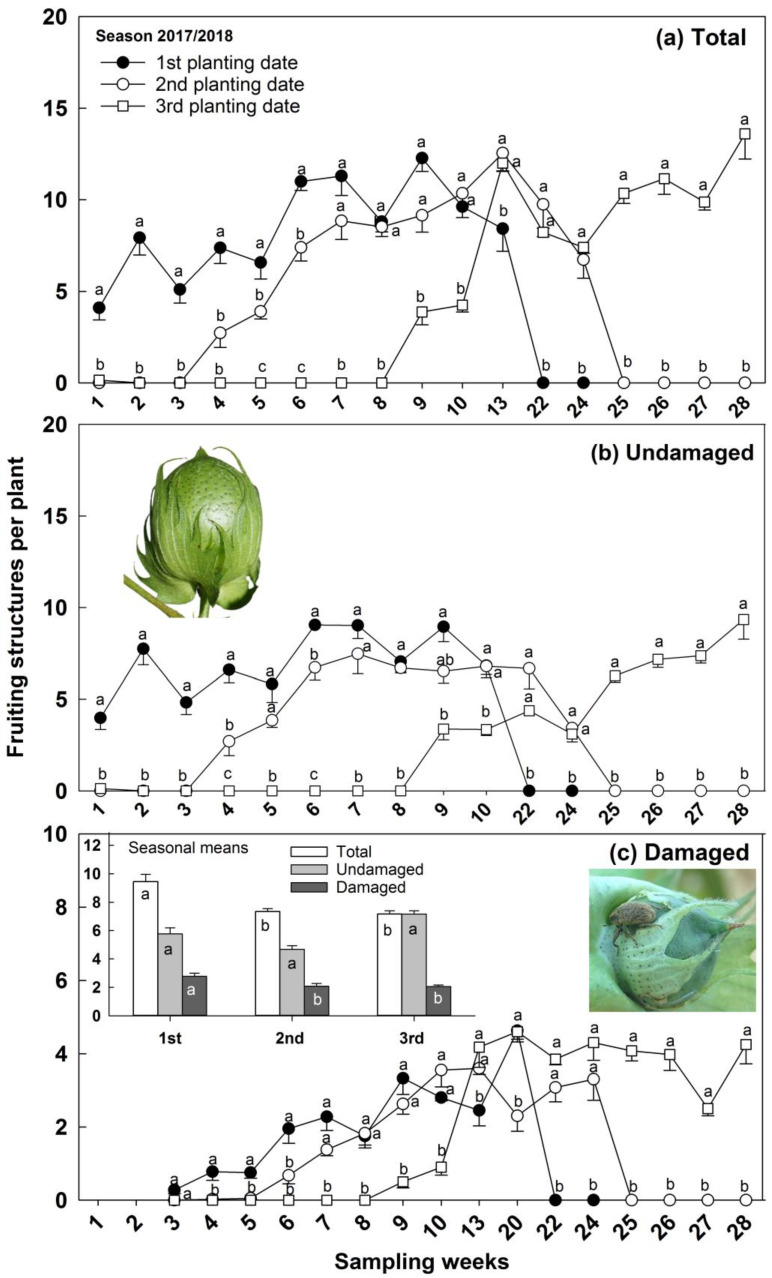
Number of cotton fruiting structures (mean -SE) per plant sampled over the season as a function of sampling weeks or seasonal mean (graphic inside) in 2017/2018 season. Symbols with different letters indicate statistical differences among planting dates within each sampling week, and the bar bearing different letters indicates differences for each fruiting structure category among sampling dates by Tukey HSD test (α = 0.05). Note: scale in damaged structures is different to better display the outcome.

**Table 1 insects-14-00599-t001:** Reproductive phenological time (mean ± SE, in days) based on the first structure observed on plants as a function of planting dates in two seasons (2014/2015 and 2017/2018).

Reproductive Structures	Planting Dates			Statistics, F*^p^*^-Value^
	20 November 2014	16 December 2014	20 January 2015	df = 2, 8
Flower bud	51.5 ± 1.44 a	39.5 ± 0.50 b	54.2 ± 0.75 a	F = 61.65^0.0001^
White flower	79.7 ± 1.25 a	67.0 ± 1.00 b	76.5 ± 0.29 a	F = 44.45^0.0001^
Young boll	81.7 ± 1.25 a	69.0 ± 1.00 b	78.2 ± 0.25 a	F = 44.18^0.0001^
Open boll	172.5 ± 0.29 a	149.7 ± 1.44 c	167.7 ± 0.25 b	F = 232.37^0.0001^
	**21 November 2017**	**20 December 2017**	**22 January 2018**	**df = 2, 9**
Flower bud	54.3 ± 0.98 a	43.3 ± 0.58 b	57.6 ± 1.02 a	F = 71.50^0.0001^
White flower	84.4 ± 0.49 a	79.0 ± 0.76 b	84.2 ± 1.11 a	F =13.96^0.0017^
Young boll	92.5 ± 0.80 ab	89.6 ± 1.06 b	97.2 ± 1.88 a	F = 8.44^0.0086^
Open Boll	197.9 ± 1.80 b	193.9 ± 0.54 b	206.8 ± 0.65 a	F = 33.16^0.0001^

Means followed by the same letter within a row do not differ by Tukey HSD test (α = 0.05).

## Data Availability

The datasets and analysis protocols used during the current study are available from the corresponding author on request.

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
