# Peer review of "Planting Date of Cotton in the Brazilian Cerrado Drives Boll Weevil (Coleoptera: Curculionidae) Infestation"

_insects, 2023, doi:10.3390/insects14070599_

Round 1
Reviewer 1 Report
This paper needs some clarification on your methodology and at least one additional year of data to make any sound conclusions. What was the protocol for deciding what ten plants were sampled within plots? You give a definition of 5mm diameter for buds, but there is no definition of what was considered a young boll. This should be defined. Your data for 2014/2015 season indicates a period approximately 2 days from finding white flowers to young bolls. In the 2017/2018 season it is approximately 10 days from white flowers to young bolls. What is the explanation for the difference (2 days versus 10 days) in development from white flower to young boll between the 2 seasons? Where and how were abscised reproductive structures maintained? Were they kept in a growth chamber at a constant temperature?
Your conclusion that "early-planted cotton is less infested than late-planted cotton" does hold true when looking at your data from the 2017/2018 growing season. However, this does not appear to be true for the 2014/2015 season where the first planting date is above ET on the first sample date and appears to stay above ET throughout the remainder of the season, while the 2nd and 3rd planting dates start out below ET (Figure 1). Your seasonal means from both seasons also indicate significantly more damaged reproductive structures in the first planting date (Figures 3 and 4). Based on this data, later planted cotton appears to have less boll weevil damage. Another year of data would be helpful to give some clarification as to if one of the presented growing seasons was an anomaly.
Grammar needs improvement. Some sentences are difficult to follow and understand what is trying to be communicated. Several instances of use of the wrong word (refuge not refugee; exposure not exposition in line 87). The discussion section is particularly hard to follow. This section needs to be more clearly communicated.
Author Response
This paper needs some clarification on your methodology and at least one additional year of data to make any sound conclusions.
Answer: All reputable scientific journals demand two years of field trials to validate the results. Long series of data are welcome as well. However, most of the time they are collected from nature not from manmade systems. We need to understand that cotton is a highly costly crop to produce because of its long developmental time, almost a half year, which requires a lot of effort and cash to run and repeat such large field trials. The results found herein are consistent across seasons and were obtained from sampling the plants during 33 and 29 weeks, a considerable amount of time. An explanation of the variation in the initial infestation found in cotton plants cultivated on the first planting date across seasons was added to the manuscript (lines 409-421) and this does not make the results obtained invalid.
What was the protocol for deciding what ten plants were sampled within plots?
Answer: The plants were randonmly selected on the central parts of the plots. The similar procedure has been adopted in the literature cited in the manuscript and enlisted in the reference list, including a recently published manuscript in Crop Protection (reference number 36). Similar protocols can be found in other scientific manuscripts cited (please, refer to the manuscripts cited in the reference list). An adjustment was added to the the line 159.
You give a definition of 5mm diameter for buds, but there is no definition of what was considered a young boll. This should be defined.
Answer: Information added, line 79. There is a lack of clear cut on SIZE because it depends on the cultivar, climatic conditions, etc. The proper approach then, is to set it based on the phenology of the fruiting structure. For example (Figure 21, page 14) http://cotton.tamu.edu/General%20Production/Georgia%20Cotton%20Growth%20and%20Development%20B1252-1.pdf
Your data for 2014/2015 season indicates a period approximately 2 days from finding white flowers to young bolls. In the 2017/2018 season it is approximately 10 days from white flowers to young bolls. What is the explanation for the difference (2 days versus 10 days) in development from white flower to young boll between the 2 seasons?
Answer: Cultivars requirements – since they were different across seasons, as well as environmental variation in the specific period (days) that these changes were occurring. The key factor though, is that pattern, which is statistically based, was consistent across seasons, no matter the numerical differences.
Where and how were abscised reproductive structures maintained? Were they kept in a growth chamber at a constant temperature?
Answer: Included in line 172. There was no justification for keeping them inside growth chambers since we would like to anticipate the emergence of the insects upon the environmental prevailing conditions. The only reason why we took them out of the field and brought them into the Lab was to keep track of and count the number of insects emerging. So, the conditions should be as close to the field conditions as possible.
Your conclusion that "early-planted cotton is less infested than late-planted cotton" does hold true when looking at your data from the 2017/2018 growing season. However, this does not appear to be true for the 2014/2015 season where the first planting date is above ET on the first sample date and appears to stay above ET throughout the remainder of the season, while the 2nd and 3rd planting dates start out below ET (Figure 1). Your seasonal means from both seasons also indicate significantly more damaged reproductive structures in the first planting date (Figures 3 and 4). Based on this data, later planted cotton appears to have less boll weevil damage. Another year of data would be helpful to give some clarification as to if one of the presented growing seasons was an anomaly.
Answer: The conclusion was adjusted in order to reflect the results found more precisely.
Reviewer 2 Report
This manuscript investigated the effects of planting date of cotton in the Brazilian Cerrado on boll weevil infestation, and concluded that planting dates in December should be preferred over the other two tested date. These results provided a new strategy to control the pest and reduce insecticide applications.
Comments:
1. The climate conditions during the two seasons of field experiments should be described.
2. L118, P2O5, the numbers should be subscripted.
3. The results section should be written more succinctly.
Author Response
This manuscript investigated the effects of planting date of cotton in the Brazilian Cerrado on boll weevil infestation, and concluded that planting dates in December should be preferred over the other two tested date. These results provided a new strategy to control the pest and reduce insecticide applications.
Comments:
1.The climate conditions during the two seasons of field experiments should be described.
Action Taken: Done.
2.L118, P2O5, the numbers should be subscripted.
Action Taken: Done.
3.The results section should be written more succinctly.
Action Taken: It was maintained as it stands now since it precisely describes all the results presented in figures and table.
Round 2
Reviewer 2 Report
accept.